# A High-Accuracy Detection System: Based on Transfer Learning for Apical Lesions on Periapical Radiograph

**DOI:** 10.3390/bioengineering9120777

**Published:** 2022-12-06

**Authors:** Yueh Chuo, Wen-Ming Lin, Tsung-Yi Chen, Mei-Ling Chan, Yu-Sung Chang, Yan-Ru Lin, Yuan-Jin Lin, Yu-Han Shao, Chiung-An Chen, Shih-Lun Chen, Patricia Angela R. Abu

**Affiliations:** 1Department of General Dentistry, Chang Gung Memorial Hospital, Taoyuan City 33305, Taiwan; 2Department of Electronic Engineering, Chung Yuan Christian University, Taoyuan City 32023, Taiwan; 3School of Physical Educational College, Jiaying University, Meizhou City 514000, China; 4Department of Electrical Engineering and Computer Science, Chung Yuan Christian University, Chungli City 32023, Taiwan; 5Department of Electrical Engineering, Ming Chi University of Technology, New Taipei City 243303, Taiwan; 6Department of Information Systems and Computer Science, Ateneo de Manila University, Quezon City 1108, Philippines

**Keywords:** PA, CNN, tooth disease recognition, image segmentation, image preprocessing

## Abstract

Apical Lesions, one of the most common oral diseases, can be effectively detected in daily dental examinations by a periapical radiograph (PA). In the current popular endodontic treatment, most dentists spend a lot of time manually marking the lesion area. In order to reduce the burden on dentists, this paper proposes a convolutional neural network (CNN)-based regional analysis model for spical lesions for periapical radiographs. In this study, the database was provided by dentists with more than three years of practical experience, meeting the criteria for clinical practical application. The contributions of this work are (1) an advanced adaptive threshold preprocessing technique for image segmentation, which can achieve an accuracy rate of more than 96%; (2) a better and more intuitive apical lesions symptom enhancement technique; and (3) a model for apical lesions detection with an accuracy as high as 96.21%. Compared with existing state-of-the-art technology, the proposed model has improved the accuracy by more than 5%. The proposed model has successfully improved the automatic diagnosis of apical lesions. With the help of automation, dentists can focus more on technical and medical diagnoses, such as treatment, tooth cleaning, or medical communication. This proposal has been certified by the Institutional Review Board (IRB) with the certification number 202002030B0.

## 1. Introduction

X-rays have been used in medical images since 1896, and they also help doctors determine whether a patient is healthy. X-rays are widely used in dental treatments [1], such as periapical radiographs (PA), bitewing radiographs (BW), and panoramic radiographs (PANO). The PA film is important in routine dental X-ray examinations because it requires lower radiation dose exposure and could identify periapical pathology efficiently. Periapical radiographs are commonly the result of trauma, caries, or tooth wear. These conditions will bring out root canal infection if the dental treatment has no intervention, and pulp necrosis may occur [2,3]. Numerous studies have confirmed that this oral problem requires prompt and thorough treatment. Otherwise, it may lead to tooth loss and repeated inflammation [4,5,6]. Detection of the peri-apical lesion and opportune endodontic procedure intervention can treat root canal infection caused by these problems. The PA film can capture local teeth, which can effectively and quickly enable dental professionals to find local details of tooth lesions [7,8] and undertake the treatment [9]. Despite advances in treatment and access to extensive care, the prevalence of apical lesions remains high [10]. Dentists use PA to find diseased areas, but it is very time-consuming to find lesions that do not contrast well with normal areas. In addition, the dentist may become tired after long working hours, thereby ignoring subtle differences. In a day, a dentist must review hundreds of X-rays of a patient and diagnose and document the patient. However, the current method for detecting periapical noise still needs to be judged by the dentist. Dentists may inevitably make mistakes as the length and number of consultations increase.

With the continuous advancement of technology, artificial intelligence has shined in many industries, such as vehicle image recognition [11], smart city [12], product recommendation systems [13], and emotion recognition [14]. The development of medical imaging is also changing with each passing day. More and more cases have shown that the combination of artificial intelligence and medical imaging has good results, such as breast cancer (BC) [15], arrhythmia [16], and lung function prediction [17]. There are also related studies in dentistry, for example, adding machine learning to the color recognition of dentures [18] or adding deep learning to the detection of tooth decay [19]. Artificial intelligence was used to improve the apical lesion detection accuracy rate of the dental Panoramic Radiograph Identification System to about 75.53% in [20], while the accuracy rate of CT image identification is about 82% [21]. For clinical use, there is still a lot of room for improvement. Therefore, in order to reduce the workload of dentists and provide more objective data, this study proposes an automatic detection system for periapical radiographs using CNN transfer learning. The purpose is to improve the symptom enhancement technology in the existing technology and to analyze and compare the final research results. In [22], a system combining PA and CNN was proposed. The authors improved the classification accuracy of batch normalization by adjusting the parameters such as the layers of convolutional blocks in the CNN. The accuracy rate obtained was as high as 92.75%. This article will use this as the main reference. However, its image enhancement improvements are not outstanding. Therefore, in this proposal, the focus is on improving image segmentation and image enhancement. In order to increase the reference value of the data obtained from the Alexnet classification model, three different CNN models are constructed, namely Googlenet, Resnet50, and Resnet101 which are trained, tested, and verified through the same database. During this period, the parameters of each model were kept consistent by the control variable method which was convenient for subsequent comparison. The research in this paper also utilizes CNN technology for dentists to diagnose symptoms and ultimately provide patients with more effective and better adjuvant treatment. The innovations of this method are as follows:In the image cropping preprocessing part, this study adds the adaptive threshold and angle rotation technology. Compared with the existing methods, this method significantly improves the image clarity and accuracy of a single tooth image.This study proposes an advanced image enhancement technique for apical lesions. It adds raw grayscale images and Gaussian high-pass filtered images to highlight the possible lesion areas and changes the color of the possible lesion area to green. Experiments show that the accuracy of the model is improved by more than 10% which proves that the proposed method is intuitive and effective.The innovation of this work is to realize the classification of various diseases. It can simultaneously judge a variety of different types of dental diseases (such as apical lesions, fillings, etc.), and the obtained final accuracy of the model proposed in this paper is as high as 93%. AlexNet even improves the accuracy up to 96.21% which is 4% higher than the state-of-the-art in [23].

The presentation structure of this proposal is as follows: Section 2 introduces the materials and methods for apical lesion detection on periapical radiographs based on transfer learning. Section 3 mainly describes and analyzes the evaluation method of the model and the experimental results. The results of the study are discussed in Section 4. Finally, the conclusions and future prospects are given in Section 5. The purpose of this paper is to predict root apical lesions located at the base of a tooth by means of a convolutional neural network (CNN).

## 2. Materials and Methods

This study is divided into three parts: image cropping, image preprocessing and CNN training. The image cropping part extracts a single tooth which helps model training more efficiently. In addition, through a series of image preprocessing techniques, possible lesion areas can be highlighted, resulting in more accurate detection. The output of these image preprocessing steps is saved in the CNN database. The clinical images used in this research were collected by attending physicians with more than three years of experience in hospital dentistry. All clinical images utilized in this research had been approved by the Institutional Review Board Statement (permission number 202002030B0). For enhancement, the most challenging problem is that after image segmentation, there is too much noise in the original apical slice and the resected part of the lesion area. Therefore, the angle of cutting or image noise reduction became the challenge of this project. This proposal uses the same Gaussian high-pass filter as that of [23] to achieve the best noise reduction result. The flow chart of this study is shown in Figure 1.

### 2.1. Image Segmentation and Retouching

In order to build a high-precision model and conform to the judgment of dentists, this research uses a single tooth image to build a clinical image database. However, since the original image is a PA composed of about three to four separate vertical teeth, segmentation of individual teeth must be performed on the original image. In [24], the vertical cutting method of dividing the image is a very good idea. However, its target image is a BW film; there would be some flaws in the process of cutting the PA film. Therefore, the segmentation method that this proposal focuses on is improved on the basis of [24] to make the segmentation more accurate. The next step is to retouch the segmented photo by adding a technique to block non-target areas on the segmented image.

#### 2.1.1. Vertical Cutting

The core concept of vertical cutting is to calculate the sum of the horizontal pixels of the image and find the point with the smallest sum making that point the segmentation point. Before performing the calculation, the image is first converted from an RGB image to a grayscale image for easier calculation. In this research, the method of iterative thresholding in [23] is improved and adaptive thresholding is used for transformation. The feature of adaptive threshold processing is that each image behaves differently from RGB to grayscale which means that each image has its own most suitable adaptive threshold. In this proposal, the most appropriate adaptive threshold is selected for each image and image transformation is performed. Figure 2 is the image result applying the conversion method using [23], and Figure 2b is the image result of the improved adaptive threshold. From a data point of view, this method can effectively improve the accuracy of image segmentation.

This proposal summarizes the range of the sum of pixels where the appropriate threshold is located by calculating the sum of the pixels of each grayscale image. However, the calculation cannot be performed immediately after converting the image because not every interdental gap is vertical. Therefore, rotating the image is a necessary step before calculation. Additionally, image rotation is performed by rotating the image 12 degrees clockwise and 12 degrees counterclockwise [24]. After rotating the image, the 24 positions and value of the minimum pixel sum for each angle can be computed. The 24 values are then compared with the smallest value being the most suitable position and angle for segmentation. Since the number of PA image teeth in the database is at most four teeth, a maximum of four cutting lines are required. This means that the above steps need to be repeated four times. However, the number of teeth in the PA images is not always four and some images have only three. In this case, the redundant dividing line needs to be removed. Therefore, this study designed two methods to address this problem. The first method is based on the relative position of the fourth cutting line to the other cutting lines. Assuming that the distance between the fourth cutting line and the other cutting lines is less than the width of one tooth, the fourth cutting line should be removed. The second method is that if the value of the fourth row is greater than the average, it means that the fourth row is very likely not on the tooth. Thus, the fourth row can be removed by this feature. Figure 3a shows the result of finding all the dividing lines. Figure 3b is the result of removing the redundant dividing lines. When the cutting line is inclined, the image segmented by the cutting line is not rectangular. However, for the subsequent training of CNN when normalized, the target in this step is a rectangular one. Therefore, this proposal adds a green vertical line to the far right and a blue vertical line to the far left of each cutting line. Moreover, it segments the first segmented image with the leftmost position of the original image and the first green vertical line as the boundary. The second segmented image is bounded by the first blue vertical line and the second green vertical line. The third and fourth cut images are similarly divided.

#### 2.1.2. Image Masks

The inclination of the cutting line is designed to match the inclination of the teeth. However, most of the cutting lines have oblique angles which means that most segmented images will contain a small fraction of adjacent teeth as shown in Figure 4a. This leads to disturbances in the accuracy of the CNN model. In view of this, this study retouches the segmented images according to the cutting line (red line in Figure 3). It sets the mask template according to the clipping line and the mask template will be superimposed with the original clipping image. This can effectively mask the non-target area as shown in Figure 4b. The modified image will be the final output and result. The retouched image will optimally preserve the desired feature areas.

### 2.2. Enhancing Lesion

In the collected original images, the root apical lesions will be affected by factors such as shooting angle, dose, and operator, thus, the lesions are sometimes inconspicuous. In this regard, this research proposes an advanced and intuitive enhancement method that can highlight the lesions. The first work is converting the RGB image to a grayscale image using a conversion formula. It then uses the Gaussian high-pass filter to filter out the noise. In order to make the lesion more obvious, the result of the Gaussian high-pass filter is used to superimpose it back to a grayscale image. Finally, the simple enhancement technique is used to change the color of the possible lesion area to green. In this way, clinical images for efficient training of CNN models can be obtained.

#### 2.2.1. Grayscale Image

The original apical section is an RGB image. However, this is not very friendly to the subsequent image processing step. To make image processing easier and accelerate the subsequent CNN training, the first step in lesion enhancement is to convert the image from an RGB three-channel image to a grayscale single-channel image. This step generates all the points needed for the subsequent steps described by the *x*- and *y*-axes of the grayscale image and at the same time achieves a more efficient process.

#### 2.2.2. Gaussian High Pass Filter

The biggest challenge in judging symptoms is the noisy points in the image. Therefore, attribute filters that reduce these noisy points are crucial. In the existing technology, there are many different filters. How to choose the most suitable filter is the key. Gaussian filters are used in two different ways. The Gaussian low-pass filter on the other hand is used to reduce certain noise points while the Gaussian high-pass filter is used to enhance dark areas. For the purpose of preprocessing to make possible apical lesions as evident as possible, the Gaussian high-pass filter is clearly the most suitable filter. This filter is able to pass high-frequency pixels and block low-frequency terms. Marginal and possibly apical lesion areas belong to high frequencies in the frequency domain. Hence, these pixels will remain in the resulting image as shown in Figure 5. The Gaussian high-pass filter [25] can be represented by Equation (1).

**Figure 5 bioengineering-09-00777-f005:**
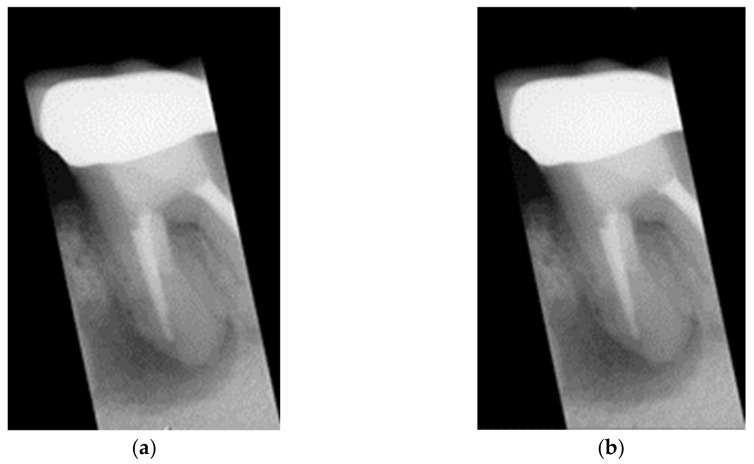
The results of the Gaussian high-pass filter. (**a**) original apical lesion image, (**b**) image after Gaussian filtering.


(1)
$$H\left(u,\;v\right)=1-e^{-D^{2}\left(u,v\right)/2D_{0}^{2}}$$


Although the image lesions after applying Gaussian high-pass filtering are obvious, the results are not as expected after model training. Therefore, other methods must be explored to enhance the lesions. In [23], in order to obtain better edges, the image preprocessing is performed by taking the grayscale image array minus the Gaussian high-pass filtered films array. The reason is that the filtered image will preserve the noisy areas and the result may not be significantly different from the input image. Therefore, by subtracting the filtered film from the input array, a clear tooth outline image can be obtained. Based on the above method, this study attempts to improve it by adding a film array to the grayscale image array after passing through a Gaussian high-pass filter to increase the likelihood of the lesion area. After applying a Gaussian high-pass filter, the filtered image retains the high-frequency pixels thus including the noise points, edges and possible lesion pixels. After superimposing this filtered image, the contrast between the light and dark areas of the grayscale original image and the filtered image becomes evident as shown in Figure 6.

#### 2.2.3. Lesion Heightened

The last step of feature enhancement is to adjust the color of the suspected lesion area where the color of the root tip is darkened. The biggest challenge for this step is how to lock the dark area of the suspected lesion. The filtered film makes it easier to find dark areas than the original image because bright areas have much larger pixel values than dark areas. However, directly using the threshold to adjust the whole block will cause other non-lesion areas in the image to be adjusted at the same time. Hence, this proposal calculates the pixel average and selects the threshold range. It can be simply reserved for possible lesions and excluded for bright areas as shown in Figure 7a. Moreover, this research used the method of calculating the pixel value difference between each pixel and its upper and lower pixels to ensure that possible lesion areas were determined. If the pixel value difference between the pixel and the pixel above or below is higher than the standard pixel value, it means that the pixel may be a point in the lesion area or just a noise point. At this time, a standard pixel value is selected. Figure 7b shows the experimental results. After using the above method of calculating the mean and calculating the difference in pixels, an AND operator is performed on the processed film to keep the points where the result is true after the AND operator. MATLAB programming provides a function to find several large regions in an image which this project uses to get the three regions that are larger than the others. Figure 7c presents the superimposed image. Finally, comparing these regions with their position in the film and their distance from the center pixel of the film, the regions that meet the above conditions will be changed into a green color and overlap with the original image. The final result is shown in Figure 7d.

### 2.3. Image Identification

In order to obtain a more scientific and reliable experimental model, the project first divides the original tooth images into a training set and a validation set according to the ratio of 4:1, as shown in Table 1. According to the transfer learning theory, the separated tooth images are cut and preprocessed according to symptoms and are then classified into the database. After that, the number of diseased teeth in the training set was expanded using horizontal and vertical mirror flips to increase the number of datasets and make it consistent with the number of normal teeth in the training set as listed in Table 2. The expanded dataset is only used to train various classification network models of CNN. It would not be used in the validation set data.

#### 2.3.1. CNN Model

In terms of deep learning, this proposal uses the tools in Matlab that support transfer learning for software development. The software environments and the hardware environments used in the proposal are listed in Table 3. To speed up the training efficiency of the CNN model, this study uses AMD R7-5800H CPU, Nvidia GeForce RTX 3070 GPU and DDR4 3200 16GB DRAM in terms of hardware performance. The architecture of each layer of the model takes AlexNet as the example of this research, as shown in Table 4. In the input stage of the model, the real training set and test set are put into the ratio of 4:1. The CNN model is trained through the classified dataset. Then, the purpose of adding a test set is to check whether the training effect deviates from the subsequent validation accuracy, thus making the experimental results more rigorous.

After deep learning, images from the validation set are randomly imported into the model. The model classifies the images according to the feature results obtained from the previous training and creates a confusion matrix by calculation to get the classification results and the accuracy of the model.

#### 2.3.2. Adjust Hyperparameter

In the training phase, the setting of hyperparameters determines the success of the model. Each parameter represents a different meaning such as the number of layers of the neural network, the loss function, the size of the convolution kernel and the learning rate. This study describes the three modified parameters, including Initial Learning Rate, Max Epoch and Mini Batch Size. In addition, the detailed information of each parameter is listed in Table 5.

*A.* 
*Initial Learning Rate*


In machine deep learning, the learning rate is a tuning parameter in the optimization algorithm. This means that the model needs an appropriate parameter which is the learning rate to get the best point of convergence. If the model has difficulty converging, it is most likely caused by the use of a too large learning rate. On the contrary, the convergence rate is too slow, which makes the model easy to overfit. Therefore, it is very important to choose an appropriate learning rate. After multiple tests on tooth images, the ideal learning rate is 0.0001.

*B.* 
*Max Epoch*


When an integrated database has passed through the CNN and has returned once, the whole process is referred to as an Epoch. However, if the Epoch is too large, it needs to be broken up into smaller pieces. With the increase of Epoch, the number of weight updates in the neural networks is also increased. The curve changed from under-fitting to over-fitting in the process of training. In general, if a CNN model has an appropriate increase in Epoch, it will lead to a better accuracy and in turn will also add training time. After repeated testing, choosing 50 as the Epoch value in each CNN model was determined by the control variable method.

*C.* 
*Mini Batch Size*


Mini Batch Size is a subset of the training set. Usually, the weights are updated and the gradient which is from the loss function is evaluated. In general, it affects the convergence of the optimization algorithm and how much memory is used in the calculation. Within a reasonable range, when the Batch Size is larger, the descending direction is more accurate and the oscillation is smaller. However, if it exceeds this range, the Batch Size is too large and local optimization or memory overflow may occur. Mini Batch Size introduces larger randomness making it difficult to achieve convergence. In this research, adjusting the approximate Mini Batch Size value to 64 can produce an ideal training result.

## 3. Results

This chapter presents the performance results of the proposed CNN model algorithm and compares it with the methods proposed in [20,23]. The proposed method for advanced symptom enhancement is also analyzed. The comparison of the image processing effect of the dataset with the results of the three CNN networks is presented for further discussion of the results.

One significant goal of this research is to enable the system to be employed in therapeutic settings. Figure 8 depicts the most common clinical workflow nowadays. Manual identification by doctors and the establishment of cumbersome medical records is a time-consuming process. The purpose of the system in this proposal is to obtain objective data for physicians prior to diagnosis and therapy after the patient takes the PA film, as shown in Figure 9.

In terms of model accuracy, this study uses the network input validation set for evaluation. The predictions obtained from the monitoring model are compared with the correct answers from the images to obtain the accuracy of the CNN. Table 6 presents the detailed training process of AlexNet and this is illustrated in Figure 10 and Figure 11. The confusion matrix and truth table according to the network model are shown in Table 7.

Figure 12 shows the training process of this paper using the symptom enhancement technique at different stages. From the results, it can be seen that when the number of iterations increases, the three curves representing different preprocessing methods all show an upward trend in accuracy. The blue line is using the Gaussian high-pass filter and discoloration at the suspected lesion, the gray line is only discoloring the lesion without using the filter, and the orange line is the no enhancement technique. The experimental results show that although all three curves show an upward trend, the results of the enhanced two curves, the blue line and the gray line, are significantly higher than the unprocessed curves. This means that preprocessing has a significant impact on the verification accuracy. In addition, the model accuracy of the technique combining the Gaussian high-pass filter with discoloration at the lesion is about 1% and 5% higher than the other two methods. These results show that the method proposed in this paper can improve the final accuracy of the model.

The technology proposed in this study is applied to clinical image judgment. Figure 13 shows the image used as the target image for clinical image judgment of symptoms. Figure 13 shows the two tooth X-rays in the red frame. The left side is the normal healthy tooth while the one on the right side is the apical diseased tooth. After implementing this technology, the classification accuracy results obtained according to the model are listed in Table 8. The accuracy of the image classification results after enhancement in this work is higher than that before disease enhancement. In clinical medicine, excellent medical quality requires high-precision judgment. The image recognition ability of CNN is exceptional. The results show that the recognition using the proposed method in this study are all above 90%.

From the research results listed in Table 9, the diagnostic accuracy of AlexNet for apical lesions can reach 96.21% which is higher than the other three models in the literature. This presents a significant improvement of more than 3% compared with 92.91% in [23] which also uses the same AlexNet architecture. Furthermore, the results of the apical lesion detection technique proposed in this paper are in stark contrast to the 75.53% accuracy reported in the tooth identification study in [20]. The research results show that the method proposed in this work is very effective and successful for apical lesions. Furthermore, it can be shown that enhancing symptoms through image preprocessing improves classification accuracy.

## 4. Discussion

In this proposal, the apical slices of multiple teeth are cut into pictures of a single tooth before training to improve the accuracy of these models. However, in the process of image cropping, this study discovered that the cutting accuracy obtained for the image by adaptive thresholding is higher than the one obtained by simple binary processing which reduces the possibility that many images contain non-target areas. The improvement of the cutting accuracy can make the effect of symptom enhancement more and thus improve the accuracy of the model. In addition, this paper uses a different method in the preprocessing of image symptoms to increase the dark area of the possible lesion area which actually helps the model accuracy to increase to more than 96%. Compared to other papers, the Gaussian high-pass filter is a tool for residual noise area to reduce noise in other projects. Changing the color of the lesion area is a different approach, and learning the features in the movie is instinctive and easy in the machine learning step. In addition, this paper proposes a hypothesis, that is the enhancement of apical lesions. The lesion area was preprocessed before importing the images into training. It can be found that the preprocessed images can further improve the recognition accuracy of CNN which is based on the premise of the quantity and quality of models and databases. The accuracy of the AlexNet model used in this research can reach up to 96.21%. Furthermore, the system’s sensitivity and specificity on clinical apical radiographs were 98.5% and 93.9%, respectively.

## 5. Conclusions

The main purpose of this study is to achieve automatic and accurate diagnosis of apical teeth, and to help dentists improve treatment efficiency. The final experimental results show that the accuracy of AlexNet can reach 96.21% which provides confidence for this project to expand the research scope, improve the accuracy and realize the clinical medical application. In the future, the research team has formulated three objectives. Firstly, the project will continue to explore the possibility of identifying multiple symptoms and achieving the classification of different symptoms. Secondly, it will try to make the model more comprehensive and improve its accuracy. Thirdly, it will develop a GUI interface integrating the functions of picture cutting, disease strengthening and disease detection which can simplify the operation process and enhance the practicability of the plan at the same time.

## Figures and Tables

**Figure 1 bioengineering-09-00777-f001:**
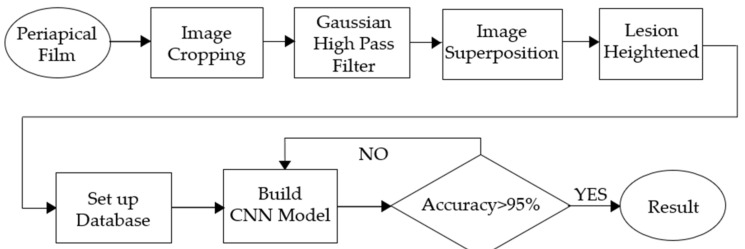
The flow chart of this research.

**Figure 2 bioengineering-09-00777-f002:**
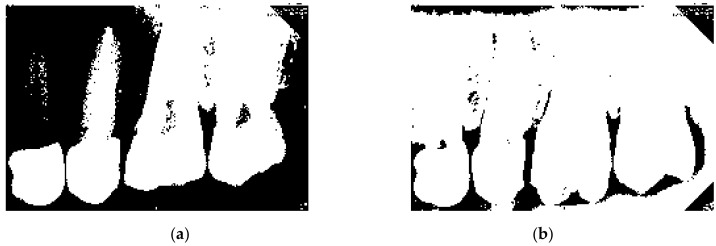
The result of the images. (**a**) image binarization, (**b**) adaptive threshold processing.

**Figure 3 bioengineering-09-00777-f003:**
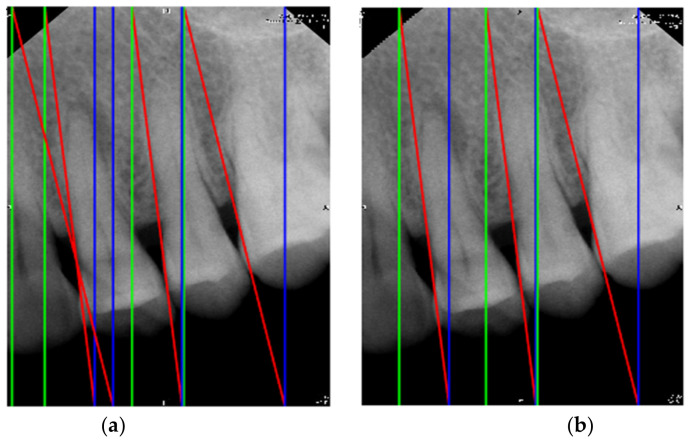
The result of cutting lines. (**a**) all cutting lines, (**b**) removed unnecessary cutting lines.

**Figure 4 bioengineering-09-00777-f004:**
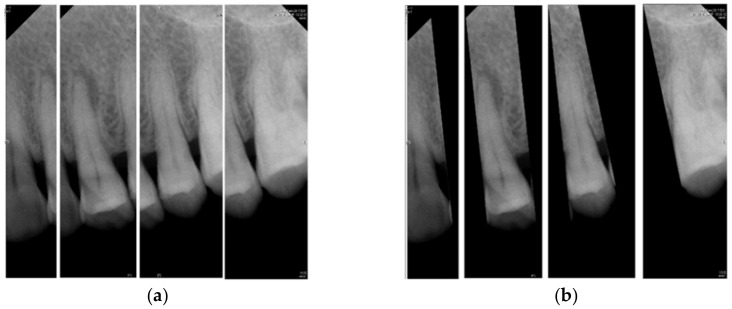
The results of the masking image. (**a**) original segmented, (**b**) retouched segmented.

**Figure 6 bioengineering-09-00777-f006:**
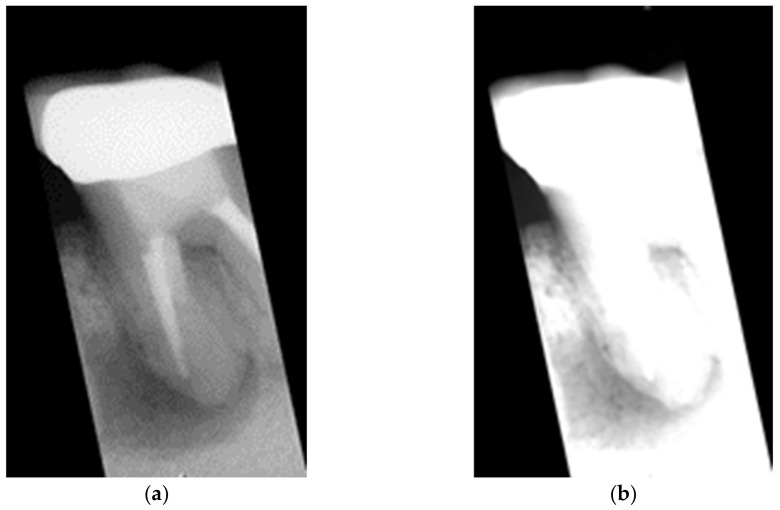
The image after adding the result of the Gaussian high-pass filter. (**a**) before, (**b**) after.

**Figure 7 bioengineering-09-00777-f007:**
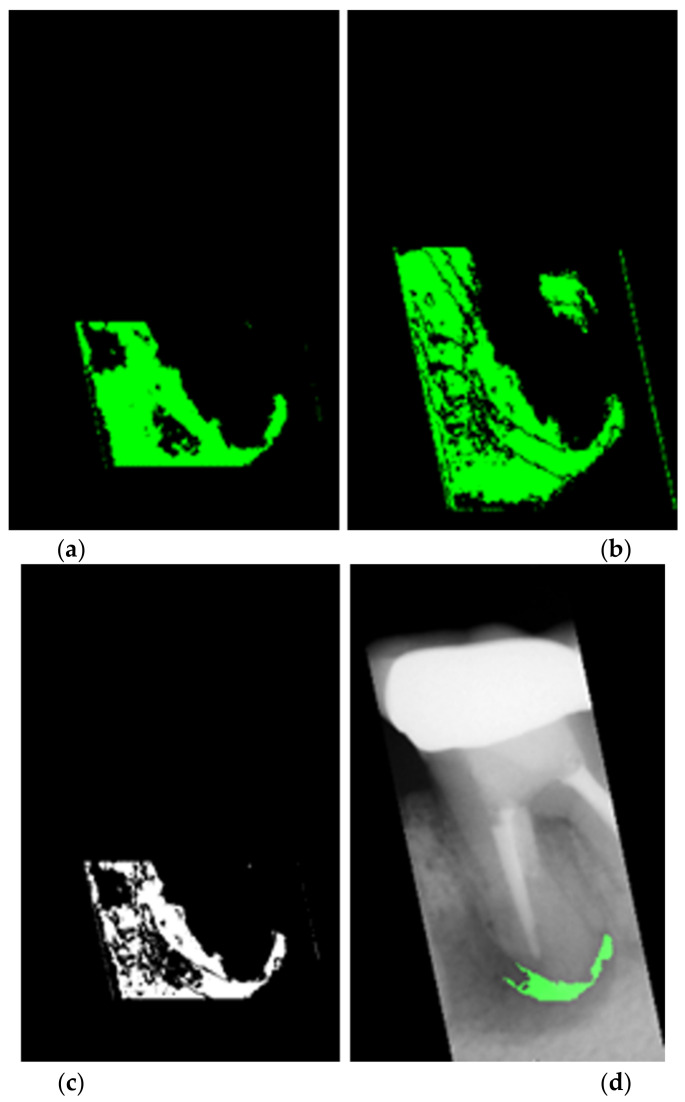
The image in different computing methods. (**a**) computed average image, (**b**) calculated pixel value difference for every pixel to the upper pixels and to the lower pixels, (**c**) superimposed image after going through the previous methods, (**d**) the final preprocessed image.

**Figure 8 bioengineering-09-00777-f008:**
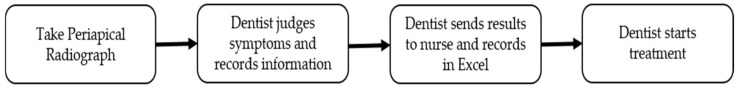
The original flow chart of clinical medicine.

**Figure 9 bioengineering-09-00777-f009:**
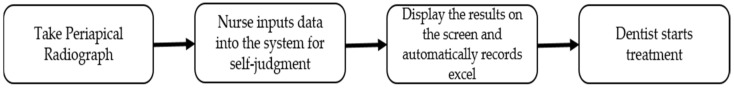
The flow chart of using this system.

**Figure 10 bioengineering-09-00777-f010:**
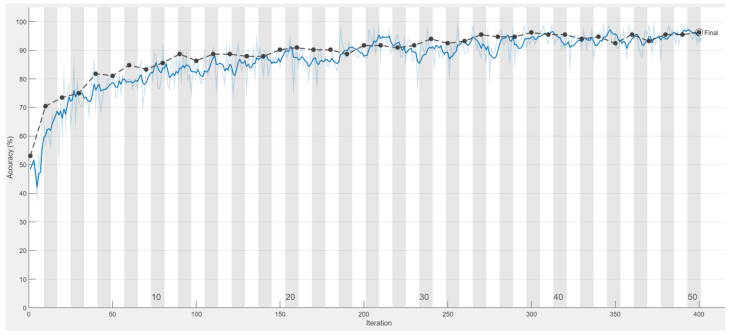
The accuracy of Alexnet model in test set which is black line and training set which is blue line during training process.

**Figure 11 bioengineering-09-00777-f011:**
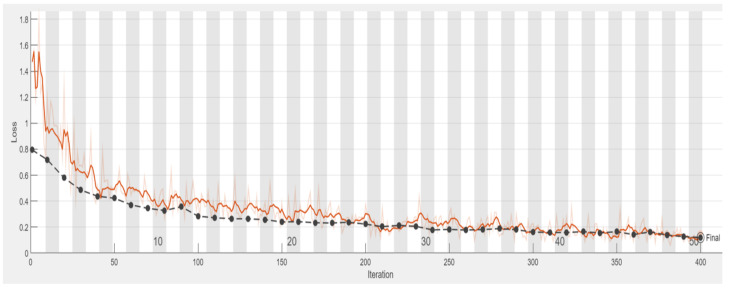
The accuracy of Alexnet model in test set (black line) and training set (orange line) during loss training process.

**Figure 12 bioengineering-09-00777-f012:**
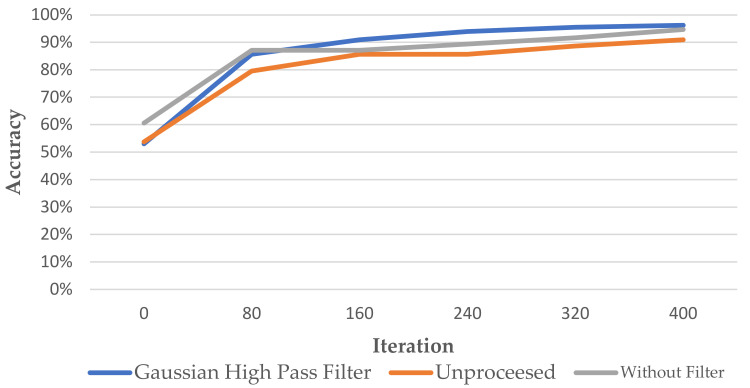
Comparison of the accuracy of AlexNet’s training process for the unprocessed image, applied Gaussian high pass filter and without filter.

**Figure 13 bioengineering-09-00777-f013:**
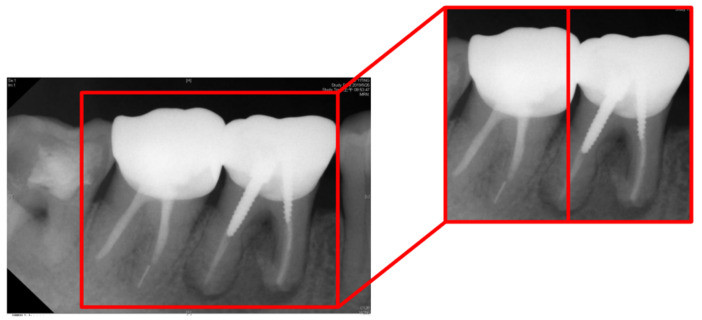
The example for validation with cropping image.

**Table 1 bioengineering-09-00777-t001:** Data Classification of the periapical image after preprocessing.

The Number of Periapical Images after Classification			
	Training Set	Validation Set	Total
Normal	332	83	415
Lesion	330 (Expanded)	15	345

**Table 2 bioengineering-09-00777-t002:** Data distribution of the original periapical image from clinical.

The Number of Original Periapical Images			
	Normal	Lesion	Total
Quantity	415	75	490

**Table 3 bioengineering-09-00777-t003:** The hardware and software platform.

Hardware Platform	Version
CPU	AMD R7-5800H
GPU	GeForce RTX 3070
DRAM	DDR4 3200 16GB
Software platform	Version
MATLAB	R2021a
Deep Network designer	14.2

**Table 4 bioengineering-09-00777-t004:** The input and output of AlexNet model.

	Type	Activations
1	Image Input	227 × 227 × 3
2	Convolution	55 × 55 × 96
3	ReLU	55 × 55 × 96
4	Cross Channel Normalization	50 × 55 × 96
5	Max pooling	27 × 27 × 96
6	Grouped Convolution	27 × 27 × 256
7	ReLU	27 × 27 × 256
8	Cross Channel Normalization	27 × 27 × 256
9	Max pooling	13 × 13 × 256
10	Convolution	13 × 13 × 384
11	ReLU	13 × 13 × 384
12	Grouped Convolution	13 × 13 × 384
13	ReLU	13 × 13 × 384
14	Grouped Convolution	13 × 13 × 256
15	ReLU	13 × 13 × 256
16	Max pooling	6 × 6 × 256
17	Fully-Connected	1 × 1 × 4096
18	ReLU	1 × 1 × 4096
19	Dropout	1 × 1 × 4096
20	Fully-Connected	1 × 1 × 4096
21	ReLU	1 × 1 × 4096
22	Dropout	1 × 1 × 4096
23	Fully-Connected	1 × 1 × 2
24	Softmax	1 × 1 × 2
25	Classification Output	1 × 1 × 2

**Table 5 bioengineering-09-00777-t005:** Hyperparameters in CNN model.

Hyperparameters	Value
Initial Learning Rate	0.0001
Max Epoch	50
Mini Batch Size	64
Validation Frequency	10
Learning Drop Period	3
Learning Rate Drop Factor	0.02

**Table 6 bioengineering-09-00777-t006:** AlexNet training process.

Epoch	Iteration	Time Elapsed	Mini-Batch Accuracy	Validation Accuracy	Mini-Batch Loss	Validation Loss
1	1	00:00:02	48.44%	53.03%	1.4716	0.7940
5	40	00:00:15	70.31%	81.82%	0.5114	0.4379
10	80	00:00:27	90.62%	85.61%	0.2726	0.3277
15	120	00:00:39	90.62%	88.64%	0.2668	0.2648
20	160	00:00:42	89.06%	90.91%	0.2776	0.2422
25	200	00:01:03	87.50%	91.67%	0.3722	0.2230
30	240	00:01:16	90.62%	93.94%	0.1955	0.1787
35	280	00:01:28	95.31%	95.31%	0.1313	0.1883
40	320	00:01:41	90.62%	95.45%	0.2768	0.1585
45	360	00:01:53	96.88%	95.45%	0.0896	0.1424
50	400	00:02:05	93.72%	96.21%	0.1520	0.1201

**Table 7 bioengineering-09-00777-t007:** The confusion matrix of AlexNet training result.

Target Class				
Category Name		Lesion	Normal	Subtotal
Output Class	Lesion	49.2%	3.0%	94.2%
	Normal	0.8%	47.0%	98.4%
	subtotal	98.5%	93.9%	96.2%

**Table 8 bioengineering-09-00777-t008:** Comparison of the clinical data and the resulting image.

Tooth Position in Figure 13	Left	Right
Clinical Data	Normal	Lesion
This Work Before Enhancement	90.91% Normal	94.70% Lesion
This Work After Enhancement	93.93% Normal	97.35% Lesion

**Table 9 bioengineering-09-00777-t009:** Image recognition accuracy obtained from a different CNN model.

	Method in [20]	Method in [23]	This Work			
			AlexNet	ResNet101	ResNet 50	Google Net
Accuracy	75.53%	92.91%	96.21%	94.70%	93.94%	87.88%

## Data Availability

Data sharing not applicable.

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
