# Peer review of "A High-Accuracy Detection System: Based on Transfer Learning for Apical Lesions on Periapical Radiograph"

_bioengineering, 2022, doi:10.3390/bioengineering9120777_

Round 1
Reviewer 1 Report
In the study of A High-Accuracy Detection System: Based on Transfer Learn- 2
ing for Apical Lesions on Periapical Radiograph by Chuo et ak-l. a novel software aiming to detect the apical lesions automatically is proposed. The study is interesting and may have important implications in the clinical dentistry. The manuscript is a quiet technical one as a dentist ı would like to see a flow diagram of the system for the clinical relevance.
The rest of the manuscript is suitable for software engineers.
Author Response
We appreciate the reviewer for providing many constructive comments and valuable suggestions, which have helped us to improve the manuscript. The responses and corrections are explained after each comment.

Reviewer 2 Report
The aim of this paper is to present a model of automatic diagnosis of apical lesions.
Below please find some suggestions from the review prior to acceptance:
1. The Introdcution part is very lengthy. Please focus on the main background why such a model was developed and describe the main aims clearly and more concise. In the current status, a clinician would not understand the relevance.
2. “The clinical images used in this research were collected by attending physicians with more than three years of experience in hospital dentistry.”
3. Please provide more details on how/why the images were chosen. Was there a pool of images and some were selected? What was the requirement for selection of the images?
4. “The IRB reviewed and determined that it is expedited review according to Case research or cases treated or diagnosed by clinical routines. However, this does not include HIV-positive cases.”
What is the rationale behind mentioning this? Is there something other with HIV-positive cases?
Author Response

(The authors gave the same response as above.)

Round 2
Reviewer 1 Report
The changes have improved the quality of the manuscript. It should be added to the discussion of the conclusion of the manuscript that the sensitivity and specificty of this detection system should be tested on a sample of clinical apical radiographs for its accuracy. This would be the next perspective for the future. research. Thank you.
Also, the use of a dental radiograph in the manuscript (displayed as a sample in figures) may require ethical committee approval and this should be concern by the authors.
Author Response

(The authors gave the same response as above.)
